# HYBRID FOURIER SCORE DISTILLATION FOR EFFICIENT ONE IMAGE TO 3D OBJECT GENERATION

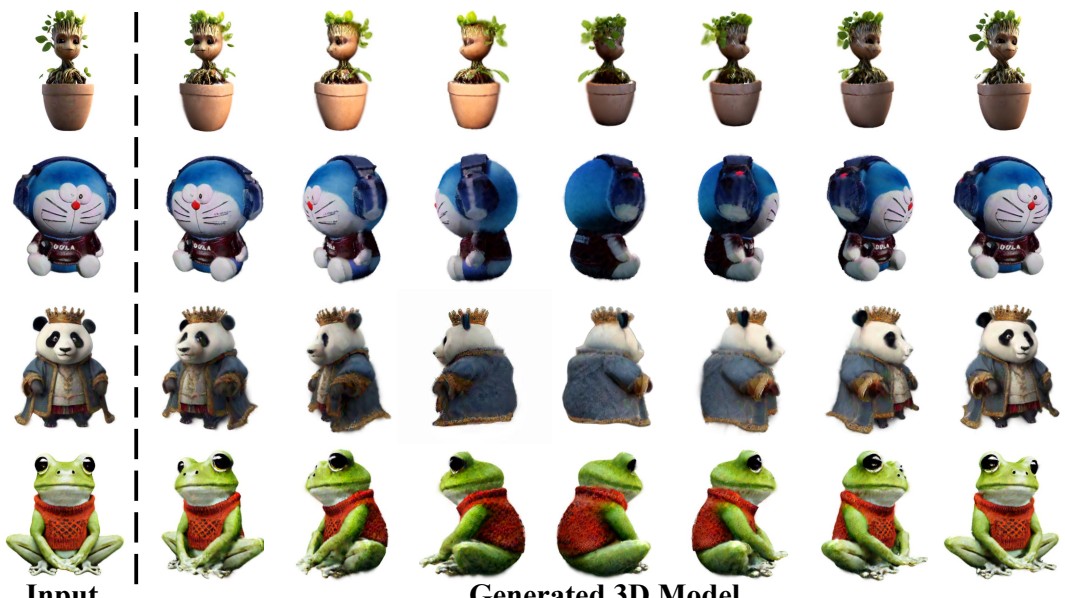

**Input**  **Generated 3D Model**

Figure 1: **Fourier123** aims at increasing the generation quality of image-to-3D task. We are able to generate a high-quality 3D object that is highly consistent with the input image within one minute.

## ABSTRACT

Single image-to-3D generation is pivotal for crafting controllable 3D assets. Given its under-constrained nature, we attempt to leverage 3D geometric priors from a novel view diffusion model and 2D appearance priors from an image generation model to guide the optimization process. We note that there is a disparity between the generation priors of these two diffusion models, leading to their different appearance outputs. Specifically, image generation models tend to deliver more detailed visuals, whereas novel view models produce consistent yet over-smooth results across different views. Directly combining them leads to suboptimal effects due to their appearance conflicts. Hence, we propose a 2D-3D **hy**brid **F**ourier **S**core **D**istillation objective function, **hy-FSD**. It optimizes 3D Gaussians using 3D priors in spatial domain to ensure geometric consistency, while exploiting 2D priors in the frequency domain through Fourier transform for better visual quality. hy-FSD can be integrated into existing 3D generation methods and produce significant performance gains. With this technique, we further develop an image-to-3D generation pipeline to create high-quality 3D objects within one minute, named **Fourier123**. Extensive experiments demonstrate that Fourier123 excels in efficient generation with rapid convergence speed and visually-friendly generation results.

## 1 INTRODUCTION

One image-to-3D generation is the process of producing exquisite high-fidelity 3D assets from a given image, which offers substantial advantages for empowering nonprofessional users to engage in 3D

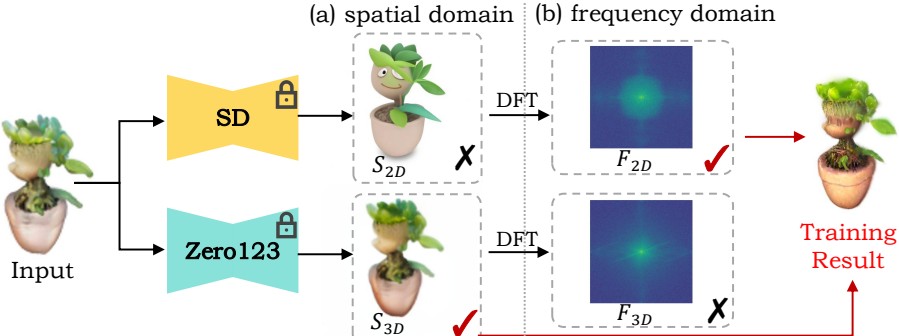

Figure 2: **Frequency analysis of Stable Diffusion (SD) and Zero-1-to-3 (Zero123)**. Discrete Fourier Transform (DFT) converts the results of SD ($S_{2D}$) and Zero123 ($S_{3D}$) to frequency domain and here we visualize their amplitude components. In the upper row, $S_{2D}$ exhibits high visual quality but distorts content structure. In the lower row, $S_{3D}$ matches the input but is over-smooth. Their frequency amplitudes, $F_{3D}$ and $F_{2D}$, are also different. We train with $S_{3D}$ for its fidelity, and $F_{2D}$ for finer details. More details can be found in Fig. 3.

asset creation. However, due to the lack of constraints, this task remains challenging despite decades of development (Ranjan et al., 2018; Wang et al., 2018; Hanocka et al., 2020). Recent advances in deep learning-based generative models (Goodfellow et al., 2014; Ho et al., 2020) have inspired an increasing number of 3D generation methods and achieved State-Of-The-Art (SOTA) effects (Hong et al., 2024; Wang et al., 2023), which are mainly divided into three strategies: 1) optimization-based 2D lifting methods, 2) novel view generation diffusion models, and 3) 3D native methods.

As existing image generation methods (Ho et al., 2020; Rombach et al., 2022) have been able to produce exquisite images, optimization-based methods attempt to using powerful 2D image generation models to achieve 3D generation (Lin et al., 2023; Wang et al., 2023). However, on the one hand, these optimization methods require tedious time for training. On the other hand, existing optimization strategy still cannot fully ustilizing the generation ability of 2D diffusion models in 3D generation, and remain some inherent problems such as Janus problem, leading to limited quality 3D generation. More recently, novel view diffusion models (Liu et al., 2023; Wang & Shi, 2023; Shi et al., 2024) and 3D native models (Nichol et al., 2022; Hong et al., 2024) that directly generate 3D objects are proposed to generate multi-view images or 3D assets within seconds. However, they are mainly trained or fine-tuned on 3D object datasets, which makes their generation ability relatively weak compared with image generation models, restricting the quality of 3D generation. To this end, we expect to develop an efficient score distillation function so that 3D generation can fully benefit from the leading generation capabilities of image generation methods for better effects.

**Motivation. (1) 2D SDS-based 3D generation**. With the powerful generation capability of 2D Stable Diffusion (SD) (Rombach et al., 2022), DreamFusion (Poole et al., 2023) utilizes SD to generate pseudo-GT views for training NeRF to achieve text-to-3D. However, SD is a 2D generative model that struggles to ensure good multi-view consistency, so a high Classifier-Free Guidance (CFG) value (Ho & Salimans, 2021) is necessary but it decreases generation quality. Additionally, SD tends to generate forward-facing images (Liu et al., 2023), which is known as the Janus problem. These limitations restrict the ability to use SD for 3D generation directly. **(2) 3D SDS-based 3D generation**. Zero123 (Liu et al., 2023) fine-tunes the pre-trained 2D SD on 3D object-level datasets, improving its multi-view consistency and 3D geometry. Similarly, DreamGaussian utilizes Zero123 to generate multi-view pseudo-GT images, training a Gaussian-based 3D representation, resulting in a better one-image 3D generation. However, Zero123 is fine-tuned on limited-scale 3D datasets, inevitably reducing its generation capability and resulting in limited generation quality. **(3) 2D SDS and 3D SDS combined 3D generation**. Intuitively, Magic123 (Qian et al., 2024) proposes a 3D generation method that combines the generative capabilities of SD with the multi-view consistency of Zero123, achieving impressive results. However, it still inherits the limitations of SD (*e.g.*, the Janus problem and content distortion) and requires a time-consuming iterative textual inversion to fit the learnable text prompt to the given image, significantly restricting the generative capacity of this text-to-image (T2I) model. Therefore, our motivation is raised:

*How to avoid the limitations of using 2D SD, a text-to-image model, for image-to-3D generation and fully unleash its generative capability?*

**Insight.** High-quality 3D generation requires three conditions: multi-view consistent geometry, clear textural details, and content that matches the input image. **(1) Spatial Domain**: As shown in Fig. 2 (a), we visualize the generation results of SD ($S_{2D}$) and Zero123 ($S_{3D}$) in the spatial (RGB) domain. Although the generated image by Zero123 ensures reasonable 3D structure and content matching with the input, it is overly smooth and lacks details. Conversely, the generated image by SD exhibits clear textures but alters content, leading to mismatches with the input image. Therefore, combining SD and Zero123 in the spatial domain is not the optimal solution. **(2) Frequency Domain**: We further transform the images to the frequency domain using Discrete Fourier Transform (DFT), as shown in Fig. 2 (b). The result of SD ($F_{2D}$) exhibits more mid-to-high frequency components (frequency increases outwards) compared to the result of Zero123 ($F_{3D}$). According to the principle of Fourier transform, higher frequencies represent finer textures. Consequently, the generation result of SD maintains its advantage of detailed textures in the frequency domain, and since it is not in the spatial domain, it does not forcibly constrain the RGB content, but only the degree of texture details. Based on the above analysis, we propose our perspective on one-image 3D generation:

*In the spatial domain, use Zero123 to ensure reasonable 3D geometry and content that matches the input. In the frequency domain, use SD to enrich the texture details.*

This solution effectively addresses the two motivations we outlined above, achieving high-quality one-image 3D generation. **(1)** Avoiding the limitations of SD: Since the Janus problem and content distortions arise in the spatial domain, we avoid using SD in the spatial domain. **(2)** Unleashing the generative power of SD: Similarly, by avoiding strong supervision on the input content in the spatial domain, we can discard the textual inversion and a high CFG value, avoiding SD fitting to local solutions and limiting the generation capability.

In summary, our main contributions are outlined as follows:

1. We use both spatial and frequency supervision for image-to-3D. The proposed hybrid Fourier Score Distillation (hy-FSD) fully unleashes the generation ability of the text-to-image model while mitigating the Janus problem and integrates its generation capabilities with the 3D priors of the novel view generation model.

2. We develop an efficient image-to-3D generation pipeline, Fourier123, utilizing hy-FSD. It enables high-quality 3D generation in one minute on a single NVIDIA 4090 GPU.

3. Extensive experiments confirm that our method significantly enhances the performance of existing optimization-based 3D generation methods, effectively produces 3D assets with reliable structure and elegant appearance.

## 2 RELATED WORK

### 2.1 3D REPRESENTATION

Recently, various 3D representation techniques have been proposed for a range of 3D tasks. Wang *et al.* (Wang et al., 2021a) employed volumetric rendering and reconstructed object surfaces by training an implicit network. Mildenhall *et al.* (Mildenhall et al., 2021) further proposed NeRF, an end-to-end model popular for enabling 3D optimization with only 2D supervision. NeRF has inspired numerous subsequent studies, including 3D reconstruction (Barron et al., 2021; Park et al., 2021; Wang et al., 2022; Li et al., 2023; Cui et al., 2024) and generation (Jain et al., 2022; Poole et al., 2023; Wang et al., 2023; Lee et al., 2024), but it consumes excessive time for optimization due to its computationally expensive forward and backward passes. Although some methods (Yu et al., 2021a; Fridovich-Keil et al., 2022; Müller et al., 2022) attempted to accelerate training, the recent developed 3D Gaussian Splatting (3DGS) (Kerbl et al., 2023) achieved real-time rendering with faster training speed, and is considered a viable alternative 3D representation to NeRF. Its efficient differentiable splatting mechanism and representation design enable fast convergence and faithful reconstruction (Luiten et al., 2023; Meng et al., 2024; Yang et al., 2024). Recent studies on 3D generation (Yi et al., 2023; Tang et al., 2024b) have adopted 3DGS to achieve faster and higher quality generation. In this work, we also employ 3DGS as the representation technique and make the first attempt to realize the optimization-based methods in both spatial and frequency domains, improving generation quality.

## 2.2 IMAGE-TO-3D GENERATION

Image-to-3D generation aims to create 3D assets from a single reference image. This problem is also known as single-view 3D reconstruction, but such reconstruction settings (Trevithick & Yang, 2021; Yu et al., 2021b; Duggal & Pathak, 2022) are limited by uncertainty modeling and often produce blurry results. Recently, diffusion models (Ho et al., 2020; Song et al., 2021) have achieved notable success in image generation, including text-to-image (T2I) (Saharia et al., 2022; Rombach et al., 2022) and novel view synthesis (Nichol et al., 2022; Liu et al., 2023; Wang & Shi, 2023; Long et al., 2023; Melas-Kyriazi et al., 2023). Several methods have attempted to extend 2D image models for 3D generation (Poole et al., 2023; Wang et al., 2023; Lee et al., 2024; Wu et al., 2024; Zhu et al., 2024), but they suffer from long optimization times as they require frequent generation of 2D images to train 3D representations. To address this, some studies explicitly injected camera parameters into 2D diffusion models for zero-shot novel view synthesis (Liu et al., 2023; Long et al., 2023). Other methods have tried to build end-to-end large reconstruction models to generate 3D assets with a single forward process (Hong et al., 2024; Tang et al., 2024a; Wang et al., 2024; Xu et al., 2024). However, the quality of the 3D assets generated by these two methods remains rough. Qian *et al.* (Qian et al., 2024) found that the T2I model (Rombach et al., 2022) has impressive 2D generation capabilities, while Zero-1-to-3 (Liu et al., 2023) tends to generate reliable 3D structures. They used both 2D and 3D priors to generate 3D. However, it still remains serious issues in tedious time costs and limited generation capacity. In this paper, we combine 2D and 3D priors from the spatial and frequency domains, respectively, fully utilizing their respective advantages and enhancing 3D generation quality.

## 3 PRELIMINARY

**3D Gaussian splatting.** We use 3DGS (Kerbl et al., 2023) as 3D representation. 3DGS uses anisotropic Gaussians to represent scenes, defined by a center position $\mu \in \mathbb{R}^3$ and a covariance matrix $\boldsymbol{\Sigma} \in \mathbb{R}^{3 \times 3}$, which is decomposed into a scaling factor $\mathbf{s} \in \mathbb{R}^3$ and a rotation factor $\mathbf{r} \in \mathbb{R}^4$. Additionally, the color of each 3D Gaussian is defined by spherical harmonic (SH) coefficients $\mathbf{h} \in \mathbb{R}^{3 \times (k+1)^2}$ for order $k$, along with an opacity value $\sigma \in \mathbb{R}$. The 3D Gaussian can be queried as:

$$\mathcal{G}(\mathbf{x}) = e^{-\frac{1}{2}(\mathbf{x}-\mu)^\top \boldsymbol{\Sigma}^{-1}(\mathbf{x}-\mu)}, \tag{1}$$

where $\mathbf{x}$ represents the position of the query point. To compute the color of each pixel, it uses a typical neural point-based rendering (Kopanas et al., 2022). Let $\mathbf{C} \in \mathbb{R}^{H \times W \times 3}$ represent the color of rendered image, where $H$ and $W$ are the height and width. Rendering process is outlined as:

$$\mathbf{C}[\mathbf{p}] = \sum_{i=1}^{N} \mathbf{c}_i \sigma_i \prod_{j=1}^{i-1}(1 - \sigma_j), \tag{2}$$

where $N$ represents the number of sampled Gaussians that overlap the pixel $\mathbf{p} = (u, v)$, and $\mathbf{c}_i$ and $\sigma_i$ denote the color and opacity of the $i$-th Gaussian, respectively.

**Latent diffusion models.** Latent Diffusion Model (LDM) (Rombach et al., 2022) consists of a pre-trained encoder $\mathcal{E}$, a denoiser U-Net $\boldsymbol{\epsilon}_\theta$, and a pre-trained decoder $\mathcal{D}$. To sample a clean image from random noise $\mathbf{x}_T \sim \mathcal{N}(\mathbf{0}, \mathbf{I})$, LDM first encodes the noise to $\mathbf{z}_T$ using the pre-trained encoder $\mathcal{E}$. Then, $\boldsymbol{\epsilon}\theta$ predicts the score function $\nabla_{\mathbf{z}_t} \log p(\mathbf{z}_t)$ to progressively remove noise, until obtaining a clean latent $\mathbf{z}_0$. Finally, the pre-trained decoder $\mathcal{D}$ is employed to decode $\mathbf{z}_0$ into the target clean image. It is evident that the main optimization objective of LDM is $\boldsymbol{\epsilon}_\theta$, which is parameterized by $\theta$. To achieve this, we first sample a clean image and encode it with $\mathcal{E}$ to obtain the Ground Truth $\mathbf{z}_0$. Then, noise of different scales is applied to it following a predefined schedule, described as follows:

$$\mathbf{z}_t = \sqrt{\bar{\alpha}_t}\mathbf{z}_0 + \sqrt{1 - \bar{\alpha}_t}\boldsymbol{\epsilon}, \tag{3}$$

where $\alpha_t \in (0, 1)$, $\bar{\alpha}_t = \prod_{i=1}^{t} \alpha_i$ and $\boldsymbol{\epsilon} \sim \mathcal{N}(\mathbf{0}, \mathbf{I})$. $\boldsymbol{\epsilon}_\theta$ is trained by minimizing the noise reconstruction loss conditioned on $\mathbf{y}$ from pre-trained language models (Radford et al., 2021):

$$\min_\theta \mathbb{E}_{\mathbf{z} \sim \mathcal{E}(\mathbf{x}), t, \boldsymbol{\epsilon}} ||\boldsymbol{\epsilon}_\theta(\mathbf{z}_t, t, \mathbf{y}) - \boldsymbol{\epsilon}||_2^2. \tag{4}$$

However, LDM lacks the ability to generate images with specified poses. To address this issue, Zero-123 (Liu et al., 2023) attempts to model a mechanism external to the camera that controls

capturing photos, thus unlocking the ability to perform new view synthesis. It uses a dataset of paired images and their relative camera extrinsics $\{\mathbf{x}^r, \mathbf{x}_{(\mathbf{R},\mathbf{T})}, \mathbf{R}, \mathbf{T}\}$ to fine-tune the denoiser as follows:

$$\min_{\theta} \mathbb{E}_{\mathbf{z} \sim \mathcal{E}(\mathbf{x}^r), t, \boldsymbol{\epsilon}} ||\boldsymbol{\epsilon}_{\theta}(\mathbf{z}_t, t, \mathcal{C}(\mathbf{R}, \mathbf{T})) - \boldsymbol{\epsilon}||_2^2, \tag{5}$$

where $\mathcal{C}(\mathbf{R}, \mathbf{T})$ is the embedding of the input view and camera extrinsics. After training, users input the image $\mathbf{x}$ and camera external parameters $\mathbf{R}, \mathbf{T}$, obtaining the target view in the appropriate pose.

**3D generation via score distillation.** Score Distillation Sampling (SDS) (Poole et al., 2023) utilizes pre-trained text-to-image diffusion models to optimize the parameters $\phi$ of a differentiable 3D representation, such as a neural radiance field or 3DGS. The loss gradient $\mathcal{L}_{\text{SDS}}$ is:

$$\nabla_{\phi} \mathcal{L}_{\text{2D-SDS}}(\phi, \mathbf{x}) = \mathbb{E}_{t, \boldsymbol{\epsilon}} \left[ w(t)(\boldsymbol{\epsilon}_{\theta}(\mathbf{z}_t, t, \mathbf{y}) - \boldsymbol{\epsilon}) \frac{\partial \mathbf{z}}{\partial \phi} \right], \tag{6}$$

where $\mathbf{x} = g(\phi, c)$ represents an image rendered from the 3D representation $\phi$ by the renderer $g$ under a specific camera pose $c$. The weighting function $w(t)$ depends on the timestep $t$, and the noise $\boldsymbol{\epsilon}$ is added to $\mathbf{z} = \mathcal{E}(\mathbf{x})$ following Eq. 3 at timestep $t$. Its key insight is to enforce the rendered image of the learnable 3D representation to adhere to the distribution of the pre-trained diffusion model.

## 4 Proposed Fourier123

In this section, we first illustrate the proposed novel score distillation sampling function, hy-FSD (hybrid Fourier Score Distillation), that supervises 3D generation in both spatial and frequency domains. hy-FSD enables the produced 3D assets to benefit from the high-quality generation capability of a text-to-image (T2I) model and the geometric prior of a novel view generation diffusion model simultaneously. Experiments validate that hy-FSD can be applied to existing 3D generation baselines for performance gains. Next, we present an efficient image-to-3D generation pipeline, Fourier123. It generates more reliable geometric structures and high-quality appearances.

### 4.1 Hybrid Fourier Score Distillation

As mentioned in Sec. 1, to fully utilize leading generation ability of SD whilst avoiding its natural limitations, we propose to use SD in frequency domain for enriching texture details of 3D assets with Fourier transform. Meanwhile, we adopt RGB results of Zero123 to ensure reasonable 3D geometry.

#### 4.1.1 Fourier Score for Stable Diffusion

The Discrete Fourier Transform (DFT), noted $\mathcal{D}$, has been widely used to analyze the frequency components of images. For multichannel color images, $\mathcal{D}$ is computed and applied independently to each channel. For simplicity, here we omit the notation related to channels. For an image $\mathbf{x} \in \mathbb{R}^{H \times W \times C}$, $\mathcal{D}$ converts it to the frequency domain as the complex component $\mathbf{X}$, expressed as:

$$\mathcal{D}(\mathbf{x})_{u,v} = \mathbf{X}_{u,v} = \frac{1}{\sqrt{HW}} \sum_{h=0}^{H-1} \sum_{w=0}^{W-1} \mathbf{x}_{h,w} e^{-j2\pi(u\frac{h}{H} + v\frac{w}{W})}. \tag{7}$$

This can be efficiently implemented with the FFT algorithm described in (Frigo & Johnson, 1998). Note that $\mathbf{X}$ contains phase and amplitude components, and the latter $\mathcal{A}(\mathbf{x})_{u,v}$ is formulated as:

$$\mathcal{A}(\mathbf{x})_{u,v} = \sqrt{\text{Re}^2(\mathbf{X}_{u,v}) + \text{Img}^2(\mathbf{X}_{u,v})}, \tag{8}$$

where $\text{Re}(\mathbf{X})$ and $\text{Img}(\mathbf{X})$ denote the real and imaginary parts of $\mathbf{X}$ respectively.

Targeting at image-to-3D generation, we employ DFT to revisit the properties of the **amplitude components** (*i.e.*, $\mathcal{A}(\mathbf{x})$), conducting frequency analysis of images generated by T2I and novel view diffusion models. As shown in Fig. 2, the frequency amplitude of results from Zero123 is concentrated in low frequencies, while SD tends to produce higher frequency results, accompanied by better subjective quality with finer details. Their discrepancy mainly lies in the middle amplitude.

Based on above observation, we employ the amplitude component of the T2I model for finer details. We focus on the frequency amplitude for two main reasons. **1)** Phase component is related to the

Figure 3: **The workflow of Fourier123**. We first use $\mathcal{F}(\cdot)$ to initialize 3D Gaussian $\phi$. $\mathcal{F}(\cdot)$ can be sphere initialization or large reconstruction model. Then, Zero123 (Liu et al., 2023) is used to supervise geometry in the spatial domain, while SD (Rombach et al., 2022) supervises appearance in the frequency domain. The whole generation process takes less than one minute.

content **structure** of the image and amplitude component means **texture** features. The images generated by SD own impressive visual quality and **fine details**, but their **structures** are deviated from the input image. **2)** Novel view diffusion model cannot provide detailed supervision since it only generates images corresponding to the input image, if input image is low-quality, Zero123 cannot improve visual quality, but SD can produce finer results. To improve quality during optimization, we have to choose amplitude component of T2I model for quality improvement. We name this optimization design as 2D Fourier score distillation and formulate it as follows:

$$\nabla_\phi \mathcal{L}_{\text{2D-FSD}}(\phi, \mathbf{x}) = \mathbb{E}_{t,\epsilon} \left[ w(t)(\mathcal{A}(\epsilon_\theta(\mathbf{z}_t, t, \mathbf{y})) - \mathcal{A}(\epsilon)) \frac{\partial \mathbf{z}}{\partial \phi} \right], \tag{9}$$

where $\mathcal{A}(\cdot)$ indicates the amplitude component in the frequency domain. Intuitively, $\nabla_\phi \mathcal{L}_{\text{2D-FSD}}$ converts the added noise $\epsilon$ and predicted noise $\epsilon_\theta(\mathbf{z}_t, t, \mathbf{y})$ into the frequency domain and calculates their differences of amplitude components, which is used to optimize 3D Gaussians $\phi$.

### 4.1.2 Hybrid Fourier Score

Although $\nabla_\phi \mathcal{L}_{\text{2D-FSD}}$ enables visual quality improvement during optimization, ensuring the generated 3D assets to be consistent with the input image is also essential. In addition, prolific structural supervision is also critical for 3D generation. To this end, we incorporate Zero123 into our distillation score. Specifically, we additionally utilize Zero123 in the spatial domain to construct 3D structure distillation sampling, expressed as:

$$\nabla_\phi \mathcal{L}_{\text{3D-SDS}}(\phi, \mathbf{x}) = \mathbb{E}_{t,\epsilon} \left[ w(t)(\epsilon_\theta(\mathbf{z}_t, t, \mathcal{C}(\mathbf{R}, \mathbf{T})) - \epsilon) \frac{\partial \mathbf{z}}{\partial \phi} \right]. \tag{10}$$

As mentioned in Eq. 5, $\mathcal{C}(\mathbf{R}, \mathbf{T})$ is the camera condition used in Zero123, which represents the camera pose of the generated novel view. By manipulating $\mathcal{C}(\mathbf{R}, \mathbf{T})$, pseudo-GTs of different views can be generated for training, leading to structural constraints.

Overall, this 2D-3D hybrid supervision, utilizing the Fourier transform, is called hybrid Fourier Score Distillation (hy-FSD), which can be expressed as:

$$\nabla_\phi \mathcal{L}_{\text{hy-FSD}}(\phi, \mathbf{x}) = \lambda_{2D} \nabla_\phi \mathcal{L}_{\text{2D-FSD}} + \lambda_{3D} \nabla_\phi \mathcal{L}_{\text{3D-SDS}}. \tag{11}$$

Note that hy-FSD can be applied to any existing optimization-based generation method, replacing their score distillation functions in a plug-and-play manner. In Sec. 5.2, we conduct such experiments on NeRF (Mildenhall et al., 2021) and 3DGS (Kerbl et al., 2023) respectively to prove our generalization and universality, proving that hy-FSD brings significant performance gains to existing methods.

### 4.2 Overall Pipeline

Our Fourier123 pipeline is simple yet effective, which generates high-quality 3D assets based on a single image within 1 minute. We employ 3D Gaussian as our 3D representation due to its superior optimization speed. Our overall framework consists of two steps: **initialization** and **optimization**. As shown in Fig. 3, we parameterize the 3D Gaussians as $\phi$. We first initialize $\phi$ and this step is formulated as $\mathcal{F}(\cdot)$. $\mathcal{F}(\cdot)$ can be implemented by sphere initialization used in Tang et al. (2024b) or other point cloud generation models (Nichol et al., 2022; Tang et al., 2024a). The latter leads to better

quality and is used in our main pipeline, but the former is still feasible. We use sphere initialization in the ablation experiments of Sec. 5.2. Next, we use Stable Diffusion (SD) (Rombach et al., 2022) and Zero-1-to-3XL (Zero123) (Liu et al., 2023) to optimize $\phi$ via the proposed hy-FSD, obtaining generated 3D assets. This process takes about 52 seconds on a single NVIDIA 4090 GPU.

**Gaussian initialization.** We employ differentiable 3D Gaussians as 3D representations. Their initialization is illustrated in Fig. 3 i), which can be modelled as: $\phi = \mathcal{F}(\boldsymbol{I})$, where $\boldsymbol{I}$ is the given input image. $\phi$ is the initialized Gaussians, which will be iteratively optimized in subsequent processes (see Fig. 3 ii)). $\mathcal{F}$ is the initialization operation, and there are typically two choices: one is to randomly initialize Gaussians within a sphere in 3D space, as DreamGaussian (Tang et al., 2024b) does. The other option is to employ a pre-trained large Gaussian generative model (*e.g.*, LGM Tang et al. (2024a)) for initialization. In our implementation, we default to using LGM for initialization. However, our method is not initialization-sensitive, meaning that random sphere initialization is also feasible. To demonstrate the generalizability of our method, we analyze the initialization configurations in Sec. 5.2 and Sec. D of the Appendix.

**Optimization with the hy-FSD.** Fourier123 uses hy-FSD to generate high-quality 3D assets with satisfactory appearances and geometries. Specifically, we use the proposed hy-FSD to optimize $\phi$. As depicted in the first term of Eq. 11, SD is employed for appearance guidance with its generations in frequency amplitude. Note that since the classifier-free guidance (Ho & Salimans, 2021) in SD is critical to its generation ability, a textual prompt is essential. This prompt can be generated by ChatGPT based on the input image, or it can be a universal text such as "*A high-quality image*". We use the latter in this paper to prove superior convenience and generalization of our method, but in fact, using texts generated by ChatGPT leads to better performance and we conduct such experiments in Sec. F of our appendix. The detailed workflow is shown in Fig. 3. We first render an image from $\phi$, then add some noise and input it to SD. SD performs a denoising process for a few steps to get 2D supervision. Both the addition and removal of noise follow the DDIM schedule (Song et al., 2021). Considering that the products of SD tend to exhibit distorted but high-quality appearances, we extract their amplitude components in the frequency domain to utilize desired appearance priors while avoiding training being misled by the distorted structure. This loss indirectly supervises 3D assets from the frequency perspective, avoiding conflicts between the generation priors of SD and other diffusion models or $\phi$ itself. Benefiting from this, we can combine Zero123 more effectively.

As depicted in the second term of Eq. 11, we use Zero123 to calculate 3D-SDS in the spatial domain, following similar operations as described in Liu et al. (2023); Qian et al. (2024). Zero123 can generate novel views that are geometrically consistent with the input image, while strictly adhering to the given input camera pose. However, it cannot improve visual quality of the input image, but SD can. Taking advantage of hy-FSD, $\phi$ is trained with rich geometric and appearance priors.

# 5 EXPERIMENT

## 5.1 IMPLEMENTATION DETAILS

**Optimization details.** We only optimize the 3D Gaussian ellipsoids for 400 iterations, where the learning rate of position information decays from $1 \times 10^{-3}$ to $2 \times 10^{-5}$. The Stable Diffusion (Rombach et al., 2022) (SD) model of V2 is selected. We set its classifier-free guidance as 7.5 because, on the one hand, a too high guidance scale (*e.g.*, 100 used in (Poole et al., 2023)) leads to low generation quality (Wang et al., 2023). On the other hand, in our method, the views generated by SD are used for supervision in the frequency domain rather than the RGB domain, which means guidance scale is not required to be too high for cross-view consistency at the pixel-level. Additionally, we leverage Zero-1-to-3XL for 3D structural supervision and its guidance scale is set to 5 according to (Liu et al., 2023). Finally, the rendering resolution is set to $512 \times 512$.

**Camera setting.** Due to the lack of pose information in the input reference image, we set its camera parameters as follows. First, we regard the reference image as being shot from the front view, *i.e.*, azimuth angle is $0°$ and polar angle is $90°$, which is similar to (Qian et al., 2024). Second, the camera is placed 1.5 meters from the content in the image, consistent with LGM (Tang et al., 2024a). Third, as we use Zero123 to supervise distillation, the field of view (FOV) of the camera is $49.1°$.

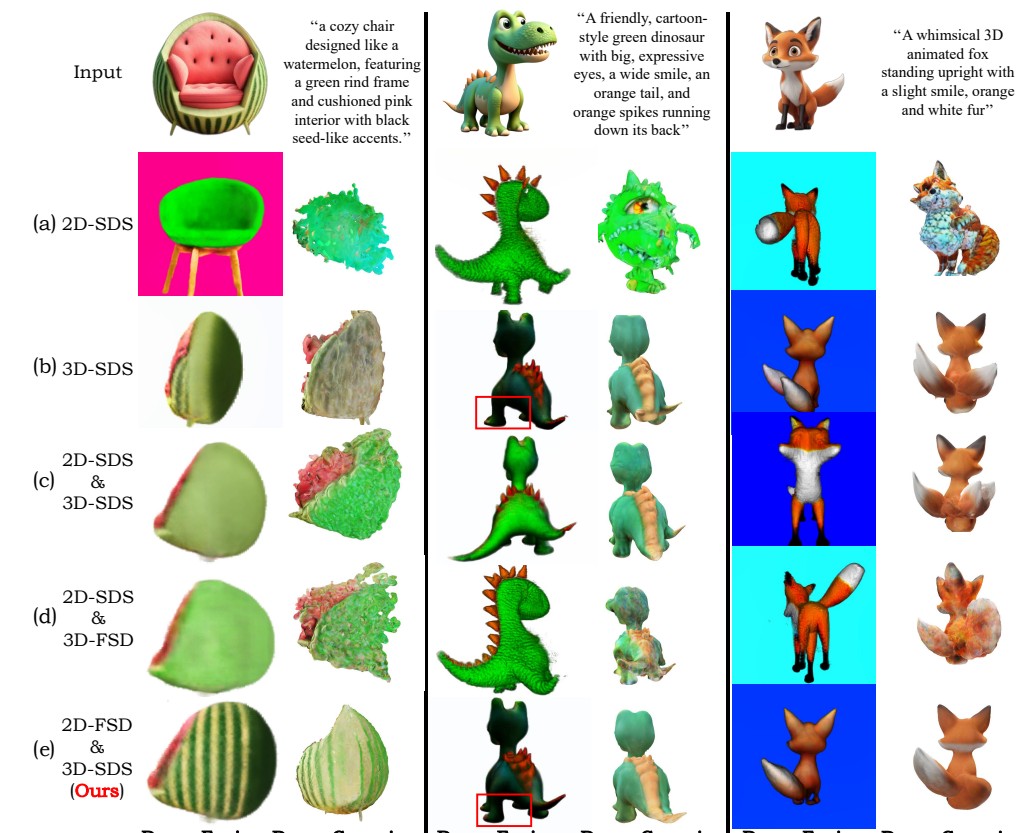

Figure 4: **Visual results of score function ablation**. We input a single image and a prompt, where the prompt is generated by ChatGPT based on the image. One can see that settings that use 2D-SDS to supervise in spatial domain all exhibit content distortion. Our hy-FSD achieves the best results.

Table 1: **Quantitative results of score function ablation**, which are measured by CLIP-similarity ↑. The best and the second best results are highlighted in **red** and **blue** respectively.

| Settings | 2D-SDS | 3D-SDS | 2D-SDS & 3D-SDS | 2D-SDS & 3D-FSD | 2D-FSD & 3D-SDS |
|---|---|---|---|---|---|
| DreamFusion | 0.6139 | **0.6950** | 0.6661 | 0.6533 | **0.7416** |
| DreamGaussian | 0.5612 | **0.6386** | 0.5923 | 0.5905 | **0.7546** |

**Evaluation metrics and datasets.** Following previous work (Poole et al., 2023; Qian et al., 2024), we report CLIP-similarity (Radford et al., 2021) as the objective metric to measure the semantic distance between rendered images and input image. Besides, we collect two types of user scores from 40 volunteers as subjective metrics, which focus on the assessment of two critical aspects in the context of image-to-3D: reference view consistency (**"User-Cons"**) and overall generation quality (**"User-Qual"**). Both are rated from 1 (worst) to 5 (best). Following (Poole et al., 2023) and (Qian et al., 2024), we conduct ablation and comparison experiments on a dataset containing 51 images, which are collected from the Objaverse (Deitke et al., 2023), OmniObject3D (Wu et al., 2023), and Internet. Moreover, 100 3D objects are randomly selected from GSO (Downs et al., 2022) to evaluate performance with lateral Ground Truth, sothat we can provide image-level metrics for more objective comparison, including PSNR, SSIM (Wang et al., 2004) and LPIPS (Zhang et al., 2018).

## 5.2 APPLYING HY-FSD TO EXISTING METHODS

To demonstrate that hy-FSD is generally beneficial to optimization-based generation methods, in this section, we apply hy-FSD to representative methods and analyze the impact of different score distillation functions on generation quality. DreamFusion (DF) (Poole et al., 2023) and DreamGaussian (DG) (Tang et al., 2024b) are chosen since they have inspired numerous subsequent works and represent two optimization-based generation methods using NeRF and 3DGS as 3D representations. Applying hy-FSD to these two classic methods effectively demonstrates its broad applicability.

Table 2: **Quantitative results of comparison experiment**. Experiments are conducted on a single NVIDIA 4090 GPU. The best and the second best results are highlighted in red and blue respectively.

| Methods | Type | CLIP-Sim ↑ | User-Cons ↑ | User-Qual ↑ | Runtime (s) ↓ |
|---|---|---|---|---|---|
| LGM (Tang et al., 2024a) | Inference-only | 0.7459 | 3.0958 | 2.8359 | **5** |
| CRM (Wang et al., 2024) | Inference-only | 0.7281 | 2.8281 | 2.8711 | 69 |
| InstantMesh (Xu et al., 2024) | Inference-only | 0.7526 | 2.7266 | 3.0664 | **11** |
| Zero-1-to-3 (Liu et al., 2023) | Optimization-based | 0.6213 | 1.3125 | 1.5039 | $2 \times 10^3$ |
| Magic123 (Qian et al., 2024) | Optimization-based | **0.7666** | 3.8203 | 2.9917 | $3 \times 10^3$ |
| DreamGaussian (Tang et al., 2024b) | Optimization-based | 0.7488 | **4.0742** | **3.1292** | 147 |
| Fourier123 (Ours) | Optimization-based | **0.8010** | **4.5251** | **3.8333** | 52 |

Table 3: **More quantitative comparison results with lateral Ground Truth**. Experiments are conducted on the GSO subset containing 100 random selected 3D objects. The best and the second best results are highlighted in red and blue respectively.

| Methods | LGM | CRM | InstantMesh | Zero-1-to-3 | Magic123 | DreamGaussian | Fourier123 (Ours) |
|---|---|---|---|---|---|---|---|
| PSNR ↑ | **17.2192** | 16.8246 | 14.8517 | 14.8944 | 14.5827 | 16.4898 | **21.5049** |
| SSIM ↑ | 0.8342 | **0.8423** | 0.7942 | 0.8314 | 0.7704 | 0.8231 | **0.8650** |
| LPIPS ↓ | **0.1806** | 0.1853 | 0.2386 | 0.1832 | 0.2764 | 0.2113 | **0.1112** |

Both DF and DG utilize SDS for 3D generation. The difference is that DF only uses SD to provide 2D priors, while DG relies solely on Zero123 for image-to-3D task. To comprehensively validate the impact of score loss functions on generation quality and the effectiveness of the proposed hy-FSD, we conduct ablation experiments with five settings: (a)**"2D-SDS"**: Vanilla DF has used 2D-SDS only. For DG, we substitute Zero123 with SD. (b)**"3D-SDS"**: Vanilla DG has used 3D-SDS only. For DF, we replace SD with Zero123. (c)**"2D-SDS & 3D-SDS"**: Following (Qian et al., 2024), we extend DF and DG to jointly use SD and Zero123 in spatial domain. (d)**"2D-SDS & 3D-FSD"**: We use 2D priors of SD in spatial domain and utilize 3D priors of Zero123 in frequency domain. (e)**"2D-FSD & 3D-SDS"** (hy-FSD): Distilling with SD in the frequency domain and Zero123 in the spatial domain.

We display visual results in Fig. 4. From the comparison between (a) and (b), one can see that (a) produces much worse structure compared with (b), which is consistent with the aforementioned statement that Zero123 can generate more suitable geometry than SD. Moreover, the structures of (c) are also worse than that of (b), such as the back views of the second and third columns. (c) even exhibits Janus problem in fox of DF, that is, producing two heads of the fox in the back view. This is because the 3D priors of Zero123 is corrupted by SD. If using SD in spatial domain but using Zero123 in frequency domain, (d) shows that 3D geometry even becomes worse. Compared with (c), the dinosaur and fox of (d) contain more distorted structures. In contrast, our method that uses SD in frequency domain and Zero123 in spatial domain, unleashes the generation ability of SD while benefiting from structure priors of Zero123. (e) performs the best visual quality than other ablation settings in both texture and structure levels.

We provide quantitative analysis using CLIP-similarity in Tab. 1 for a more objective illustration, average scores are calculated on 151 3D objects, including the collected 51 cases and 100 samples from GSO data. One can see that directly using zero123 can achieve acceptable image-to-3D effect, but with hy-FSD, we further bring notable performance gains to existing pipelines.

## 5.3 COMPARISON

As said in Sec. 4.2, text generated by ChatGPT and a universal text, "*A high-quality image*", can both be used in SD. In this section, we display our results generated with universal text to prove that we can achieve SOTA effects without elaborated texts. However, using text produced by ChatGPT in SD actually performs better. We also showcase results of this optimal setting in Sec. F of appendix.

**Qualitative comparison.** We provide qualitative comparisons of generation quality in Fig. 5. We primarily compare with three baselines from inference-only methods (Tang et al., 2024a; Wang et al., 2024; Xu et al., 2024) and three optimization-based methods (Tang et al., 2024b; Liu et al., 2023; Qian et al., 2024). In terms of generation speed, our approach exhibits significant acceleration compared to optimization-based methods. Regarding the quality of generated 3D assets, our method outperforms both inference-only and optimization-based methods, especially with respect to the fidelity of 3D geometry and visual appearance. Zero123 and DreamGaussian both only use 3D SDS and Magic123 uses 2D and 3D SDS. One can see that appearances of Zero123 and DreamGaussian tend to be smooth and blurry, which are influenced by the limited quality pseudo-GTs generated by

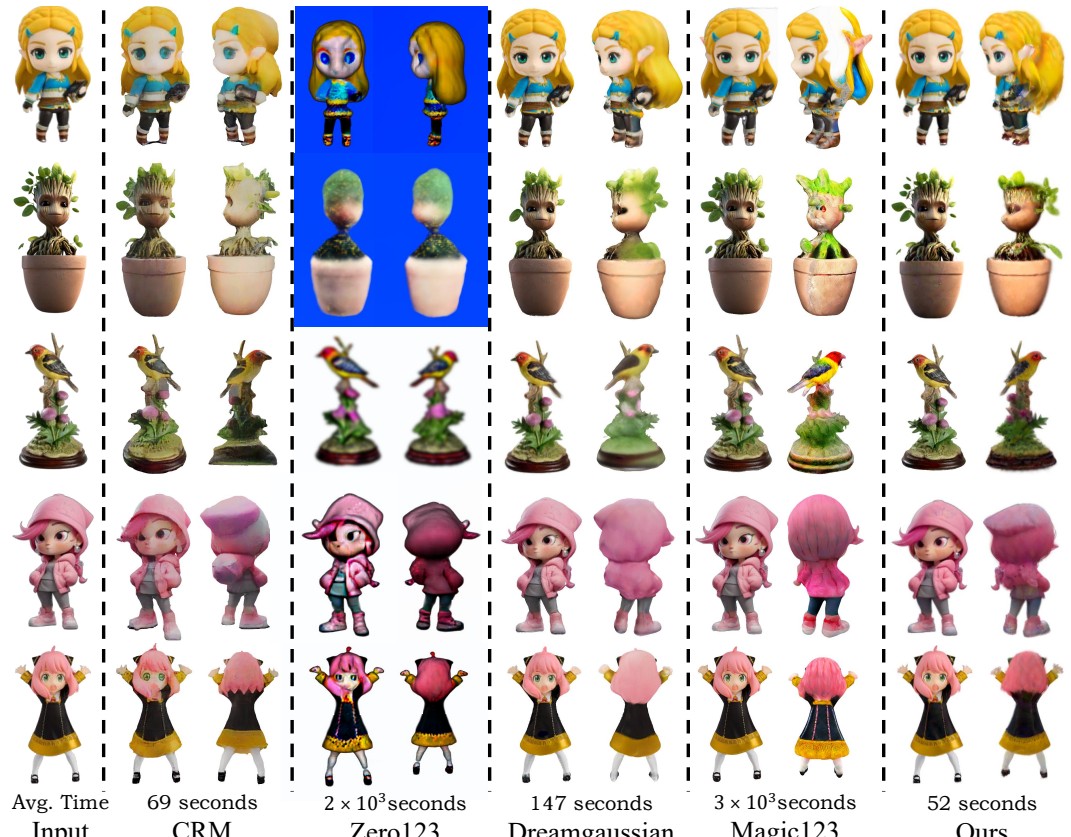

| Avg. Time | 69 seconds | $2 \times 10^3$ seconds | 147 seconds | $3 \times 10^3$ seconds | 52 seconds |
| Input | CRM | Zero123 | Dreamgaussian | Magic123 | Ours |

Figure 5: **Visual comparison**. Input images are given on the left and runtime is listed below. For clear comparison, we omit LGM and InstantMesh here and the full version can be found in Sec. A.

Zero123. Magic123 produces relatively clear results, but its geometric structures are distorted, such as the first, second, and last rows. This is because results from SD corrupts the 3D priors of Zero123. In general, Fourier123 achieves SOTA balance between appearance and structure.

**Quantitative comparison.** We first use the collected 51 3D objects to evaluate performance. In Tab. 2, we report "CLIP-similarity" and two types of user scores to measure the generation ability of different methods. The average runtime on input with the resolution of $512 \times 512$ is provided on the right to evaluate their efficiency. Note that the "Zero-1-to-3" in Tab. 2 refers to training a NeRF (Mildenhall et al., 2021) with the pseudo-GTs from different views generated by Zero-1-to-3, which shares the same training settings as in (Liu et al., 2023). One can see that our method outperforms the compared methods in terms of generation quality and achieves SOTA speed among optimization-based methods.

To enable objective image-level evaluation with lateral Ground Truth, we further use 100 3D objects from GSO dataset. Considering that both Wonder3D (Long et al., 2023) and CRM (Wang et al., 2024) selected 30 cases from GSO, we believe the subset we used is sufficient for quantitative comparison. PSNR, SSIM, and LPIPS are reported in Tab. 3. It is easy to see that our method performs superior 3D generation ability compared to existing baselines.

## 6 CONCLUSION

In this work, we present Fourier123, a 3D object generation framework that achieves high-quality image-to-3D generation. Two key contributions of our work are: 1) We propose a 2D-3D hybrid Fourier score distillation function, which attempts to fully unleash generation ability of T2I model for efficient image-to-3D generation. 2) We design a generative Gaussian splatting pipeline, called Fourier123. It can produce ready-to-use 3D assets from a single image within one minute. We believe that this method of distilling 3D assets from a frequency domain perspective provides a novel approach for the 3D generation task, and this work demonstrates its effectiveness.

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

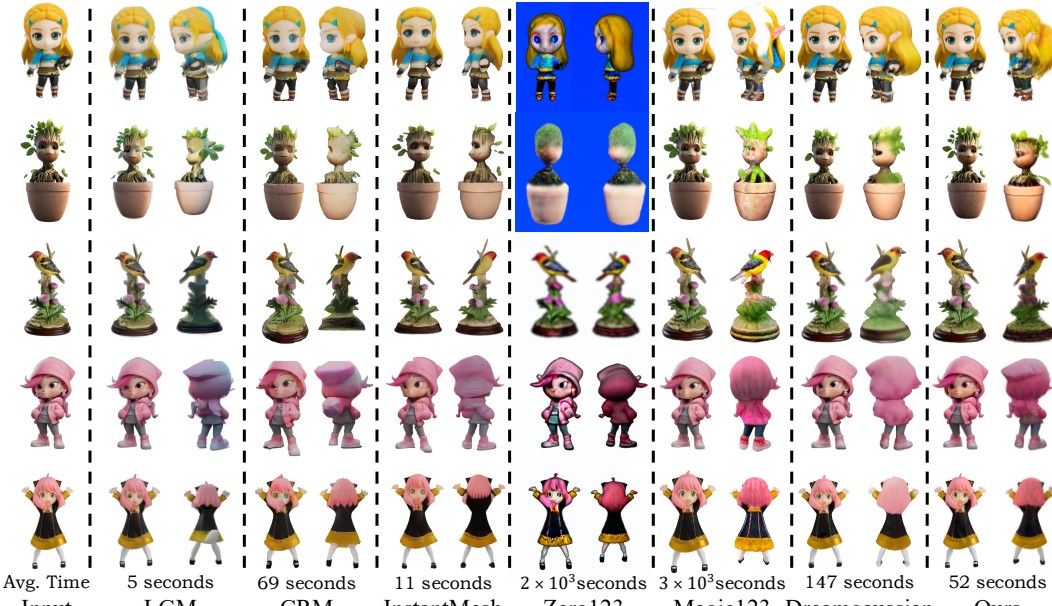

| Avg. Time | 5 seconds | 69 seconds | 11 seconds | $2 \times 10^3$ seconds | $3 \times 10^3$ seconds | 147 seconds | 52 seconds |
|---|---|---|---|---|---|---|---|
| Input | LGM | CRM | InstantMesh | Zero123 | Magic123 | Dreamgaussian | Ours |

Figure 6: **Full visual comparison**. The input images are given on the left and runtime is listed below. Our method achieves better generation quality with competitive efficiency.

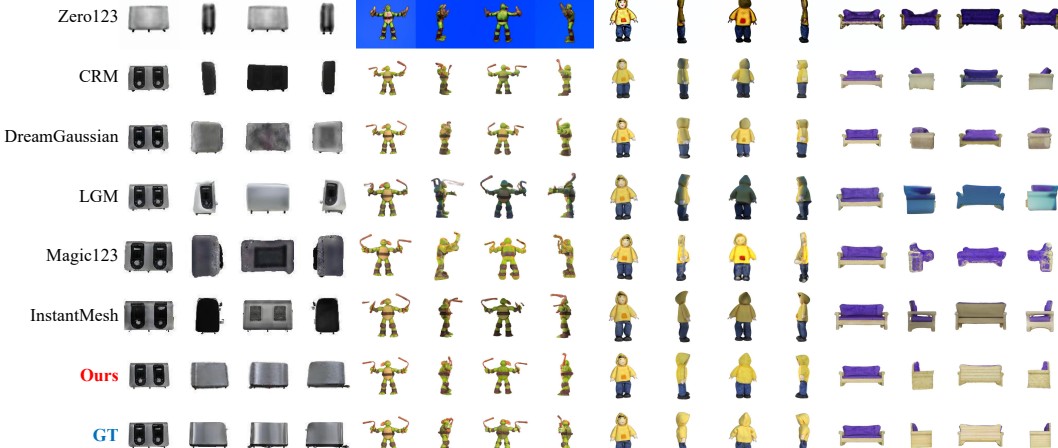

Figure 7: **Visual comparison with lateral Ground Truth**. We display results of different methods on the 100 GSO objects, attaching the corresponding lateral views of Ground Truth to compare more intuitively.

## A   MORE SUBJECTIVE COMPARISON

The subjective comparison mentioned in Sec. 5.3 is simplified due to the limited space. In Fig. 6, we report the full comparison, including results of the two inference-only methods (LGM (Tang et al., 2024a) and InstantMesh (Xu et al., 2024)). Furthermore, considering that Fig. 6 only reports the cases that are collected from Internet, that is, lacking the lateral Ground Truth, we also provide the results produced on GSO data from different methods in Fig. 7. Since GSO contains 3D models, we can render lateral Ground Truth for subjective comparison. One can see that the results produced by our method not only contain the best appearances and structures, but are also more consistent with the Ground Truth, which effectively proves the effectiveness of our method.

## B   FREQUENCY VISUALIZATION

In Fig. 8, we display more visual results from the frequency domain to illustrate the frequency differences between the outputs of Zero-1-to-3 (Zero123) and Stable Diffusion (SD). We provide

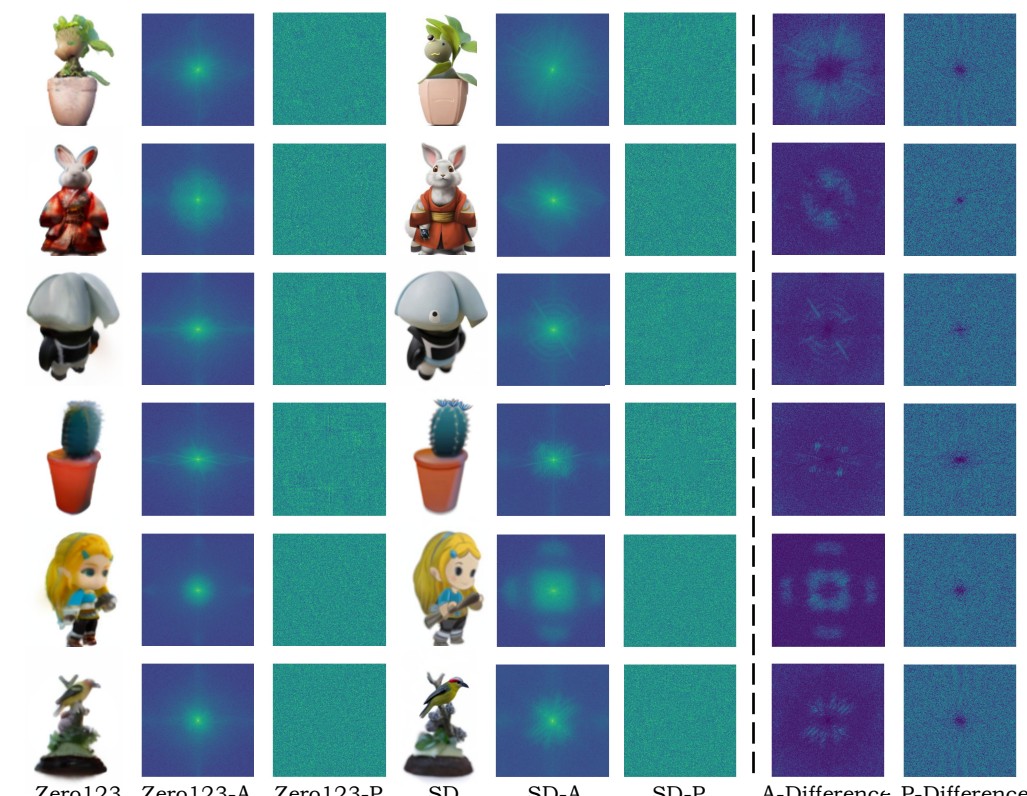

Figure 8: **Frequency difference**. We visualize the results generated by Zero123 and SD, and their Amplitude and Phase components in the frequency domain. Zero123 and SD produce very different appearances, with different frequency domain distributions. Overall, the results of SD exhibit better appearances. The amplitude difference between SD and Zero123 is clear that mainly focuses on middle frequency, however, their phase differences are irregular and meaningless.

the corresponding amplitude component of Zero123 ("Zero123-A") and SD ("SD-A"), the phase component of Zero123 ("Zero123-P") and SD ("SD-P"), and their difference, *i.e.*, "A-Difference" and "P-Difference" at the right end. One can see that in the spatial domain, novel views generated by SD exhibit higher quality compared with those of Zero123, but their structure and identity features are offset. Whilst in the frequency domain, as shown in "A-Difference" of Fig. 8, results of Zero123 and SD exhibit different amplitude distributions. We believe that the amplitude in frequency domain results of SD is more consistent with that of high-quality images, thus using the frequency amplitude of SD for finer textures. Meanwhile, "P-Difference" shows that the phase components of Zero123 and SD are very different. Considering that phase component represents the content structure of the image, and Zero123 produces cross-view geometric-consistent views at the pixel level but SD cannot, we do not use the frequency phase of SD, but employ Zero123 at the pixel level. This combination of distillation is called hy-FSD.

## C    OTHER 2D APPEARANCE SUPERVISION IN FREQUENCY DOMAIN

In hy-FSD, we adopt a 2D diffusion model for its appearance priors. However, can the diffusion model be replaced by other image enhancement methods, such as image Super-Resolution (SR), to enhance visual quality? In this section, we use two classical SR methods, *i.e.*, Bicubic and R-ESRGAN Wang et al. (2021b), to distill 3D assets in the frequency domain to validate their effectiveness.

As shown in Fig. 9, we visualize the results of diffusion models (Zero123 and SD) and SR methods (R-ESRGAN and Bicubic) in both the spatial and frequency domains. For SR methods, we render an image from the 3D Gaussians, with the same camera settings as those used in Zero123 and SD, then input the rendered image to SR methods to get their results. Obviously, the appearances enhanced by SR methods cover wider frequency levels, exhibiting a totally different distribution from that of

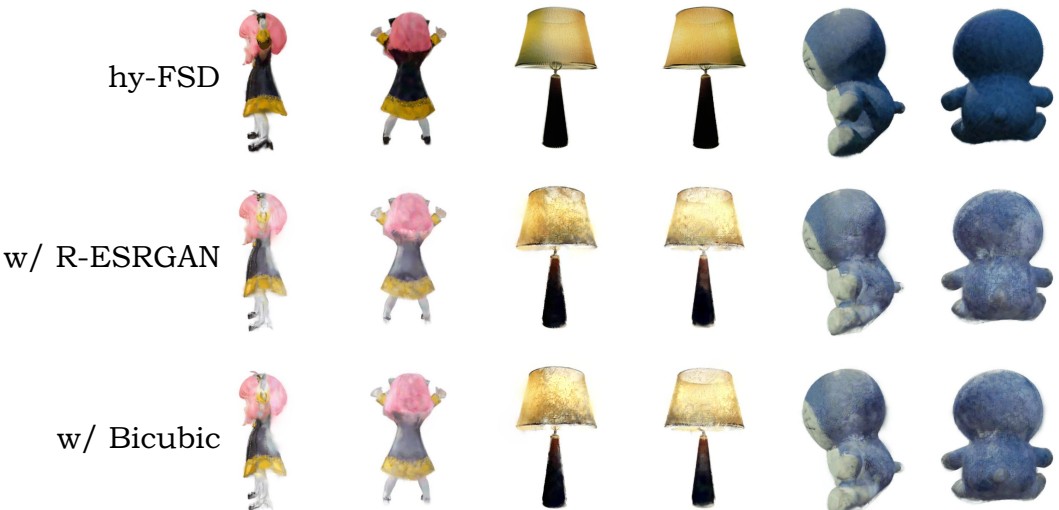

| Zero123 | Stable Diffusion | R-ESRGAN | Bicubic |

Figure 9: **Frequency results of different models**. We visualize the results generated by different models, including two diffusion models (Zero123 and SD) and two Super-Resolution (SR) methods (R-ESRGAN and Bicubic). One can see that enhanced images of SR methods cover wider frequency levels, which are very different from that of diffusion models.

hy-FSD

w/ R-ESRGAN

w/ Bicubic

Figure 10: **Results of training with super-resolution methods**. We alternate the SD model used in hy-FSD with other super-resolution methods, *i.e.*, distilling with Zero123 in spatial domain and using SR methods in frequency domain. One can see that supervising with SR methods in the frequency domain leads to worse results than training with the proposed hy-FSD.

diffusion models. Then, similar to the setting of hy-FSD, we combine the frequency results of SR methods and the spatial results of Zero123 to distill 3D assets. The final 3D assets are visualized in Fig. 10. The setting "w/ R-ESRGAN" means using Zero123 in the spatial domain and employing R-ESRGAN in the frequency domain, while the setting "w/ Bicubic" means adopting Bicubic in the frequency domain. One can see that compared to the results trained by hy-FSD (the first row), 3D assets generated by SR methods (the next two rows) exhibit worse visual quality. We believe this is because the frequency results of SR methods cover a too wide range of frequency levels, making training difficult. THIS also proves that using Stable Diffusion to supervise in the frequency domain is an appropriate choice for 3D generation.

## D ABLATION ON INITIALIZATION

To prove that the effectiveness of our method is not highly rely on LGM initialization, we use sphere initialization in Fourier123 and compare its performance with others in Tab. 4, results are also calculated on the 100 3D objects from the GSO data. Obviously, this setting still produces competitive results, that is, Fourier123 w/ Sphere outperforms 3 well-known SOTA methods: DreamGaussian, InstantMesh, and Magic123. Considering that DreamGaussian represents methods that using 3D SDS and Magic123 is the SOTA approach that uses both of 2D and 3D SDS. We believe that this

Table 4: **Ablation on Initialization**. Fourier123 can still be effective with sphere initialization. The best and the second best results are highlighted in **red** and **blue** respectively.

| Methods | InstantMesh | Zero-1-to-3 | Magic123 | DreamGaussian | Fourier123 w/ Sphere | Fourier123 w/ LGM |
|---|---|---|---|---|---|---|
| PSNR ↑ | 14.8517 | 14.8944 | 14.5827 | 16.4898 | **16.7452** | **21.5049** |
| SSIM ↑ | 0.7942 | **0.8314** | 0.7704 | 0.8231 | 0.8091 | **0.8650** |
| LPIPS ↓ | 0.2386 | **0.1832** | 0.2764 | 0.2113 | 0.2012 | **0.1112** |

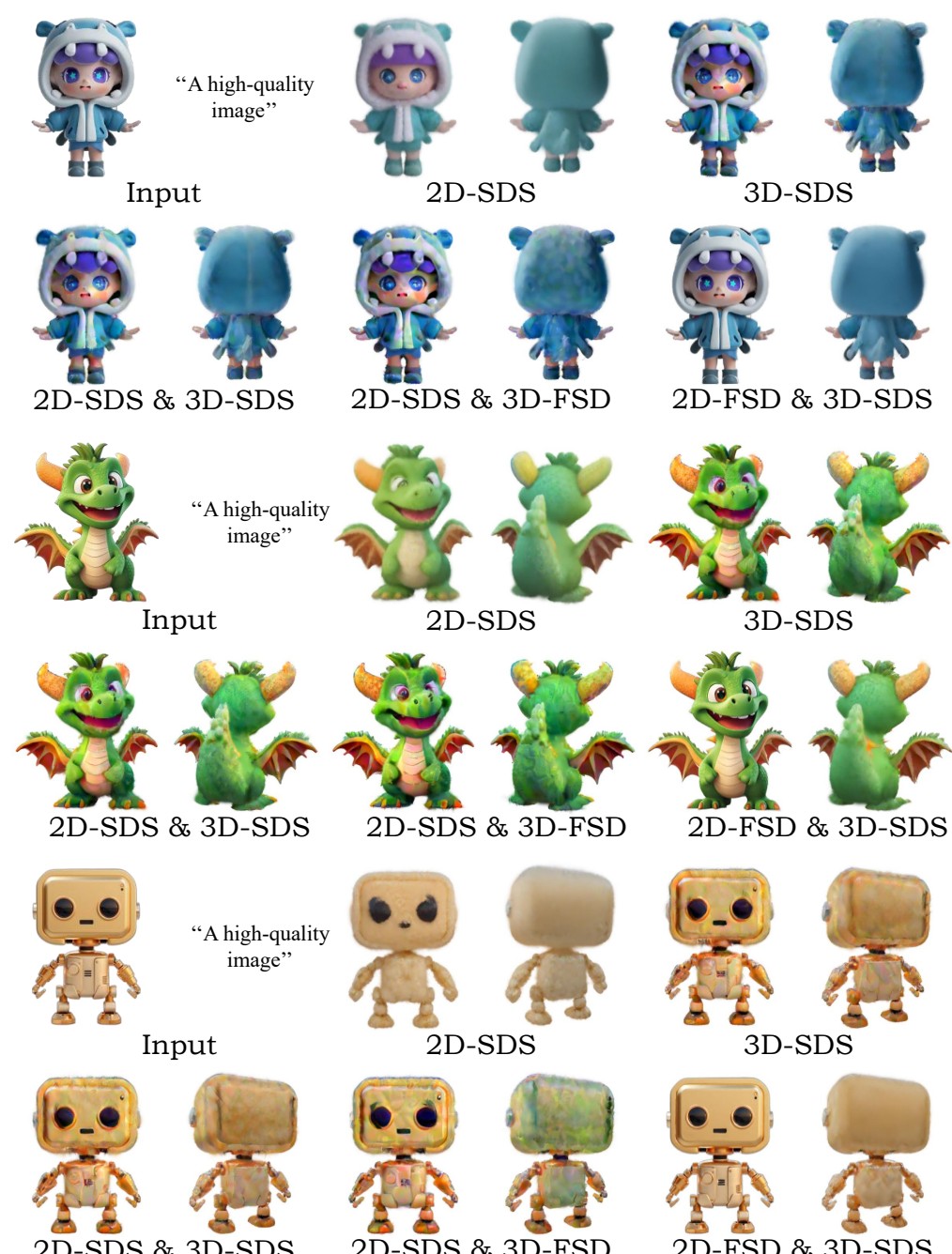

Figure 11: **Ablation study on Fourier123**. We further conduct ablation study on the proposed pipeline, Fourier123. One can see that the setting that uses hy-FSD realized the best visual results.

comparison can effectively prove the superiority of our method. More importantly, notice that DreamGaussian has to extract mesh from the generated Gaussians and refine mesh texture to get acceptable effects. But Fourier123 can be directly optimized from shpere initialization and get better 3D assets without the second stage fine-tune used in DreamGaussian, which is much better and more efficient.

Table 5: **Ablation study on Fourier123**, which are measured by CLIP-similarity ↑ and conducted on the same dataset used in the main paper. The best and the second best results are highlighted in red and blue respectively.

| Settings | 2D-SDS | 3D-SDS | 2D-SDS & 3D-SDS | 2D-SDS & 3D-FSD | 2D-FSD & 3D-SDS |
|---|---|---|---|---|---|
| Fourier123 | 0.7474 | 0.7780 | **0.7802** | 0.7642 | **0.8010** |

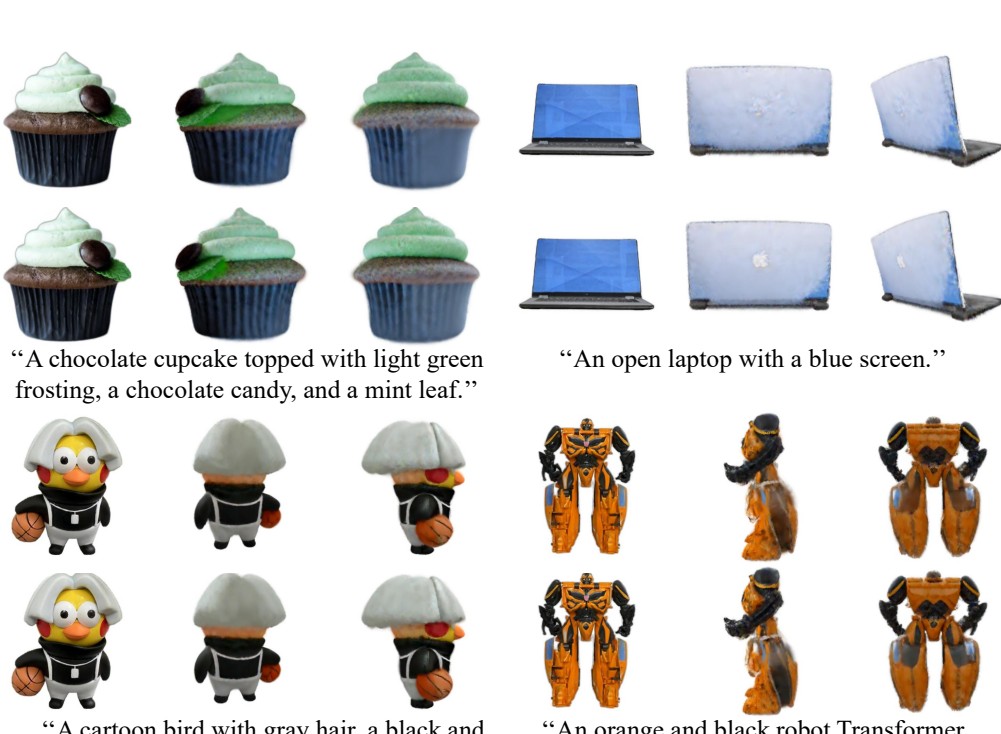

"A chocolate cupcake topped with light green frosting, a chocolate candy, and a mint leaf."

"An open laptop with a blue screen."

"A cartoon bird with gray hair, a black and white outfit, and holding a basketball."

"An orange and black robot Transformer Bumblebee."

Figure 12: **Comparison of results generated with specific text and universal text**. For each group, we showcase the results generates with a universal text, *i.e.*, "*A high-quality image*", on the upper row, and display the results generated with the specific text produced by ChatGPT on the lower row. The corresponding specific texts are given below.

# E ABLATION SCORE FUNCTION ON FOURIER123

In our main paper, we have demonstrated the effectiveness of the proposed hy-FSD by applying it to existing 3D generation baselines, including DreamFusion and DreamGaussian. In this section, we additionally conduct an ablation study on the proposed Fourier123 pipeline to analyze its performance and further demonstrate the effectiveness of hy-FSD.

As shown in Fig. 11, we train 3D Gaussians with different distillation functions. Obviously, results of the setting using the proposed hy-FSD ("2D-FSD & 3D-SDS") exhibit the best visual quality. To evaluate the performance of different settings more objectively, we further report quantitative ablation results in Tab. 5. Note that this experiment uses the same dataset employed in the main paper. Apparently, using only 2D or 3D diffusion priors in the spatial domain cannot achieve satisfactory results. Although combining both of them to supervise at the pixel level improves performance, one can see that hy-FSD brings the most significant performance gains, which is consistent with the ablation studies on DreamFusion and DreamGaussian.

Table 6: Quantitative comparison of results generated with specific and universal texts. Values are measured by CLIP-similarity ↑.

| Settings | w/ universal text | w/ specific texts |
|---|---|---|
| Fourier123 | 0.8010 | 0.8193 |

| Input | 2D-PE & 3D-SDS | 2D-FSD & 3D-Phase | 2D-FSD & 3D-SDS |

Figure 13: **Using supervisions from other feature domains**. On the one hand, we apply position embedding to results of SD rather than Fourier transform, building "2D-PE&3D-SDS". On the other hand, we only use the phase component of results from Zero123 to train. building "2D-FSD&3D-Phase". Neither of them produce 3D objects well, proving the suitability of our method.

## F  ABLATION ON THE TEXTUAL PROMPT

In Sec. 4.2 of the main paper, we claimed that the prompt used in Stable Diffusion can be generated by ChatGPT based on the input image, or can be a universal text, that is "*A high-quality image*". Here we showcase the results generated using two different prompts to support this statement.

As shown in Fig. 12, we showcase four groups of comparison. Each group consists of three views generated with universal text (the upper row) and three images generated with specific text (the lower row). One can see that the results generated by specific and universal texts are similar. The former exhibits slightly better visual quality with more natural details and textures. In Tab. 6, we report their quantitative values on the same dataset used in the main paper, CLIP-similarity is used to measure. Although the setting using universal text has achieved SOTA generation quality, training Fourier123 with the specific text performs even better. In this paper, we primarily use the suboptimal setting that employs universal text to demonstrate that we can achieve superior image-to-3D generation without well-designed prompts.

## G  EXPLORING OTHER FEATURE DOMAINS

**1)** Considering that phase component means content structure and we want to use structure priors of Zero123. In the main paper, we directly use its RGB results and build "2D-FSD&3D-SDS" to optimize 3D Gaussian. Here we supplement a study that uses the phase component of Zero123 and discard its RGB results, that it, "2D-FSD&3D-Phase". **2)** On the other hand, we want to demonstrate that the choice of frequency domain is suitable and it cannot be replaced by other feature domains. To this end, for results of SD, we attempt to apply sin-cos Position Embedding to increase its feature channels and take this feature map to optimize. The final optimization function is "2D-PE&3D-SDS".

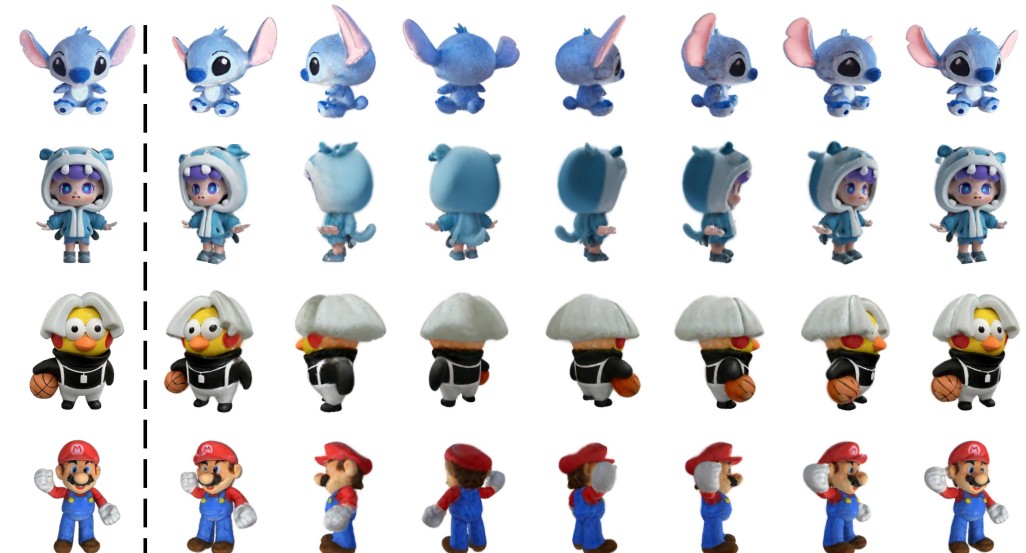

Figure 14: **More visual results of Fourier123**. We showcase part of our visual results here. All of them can be found in the HTML file in supplementary material.

Some results of the above mentioned two settings are given in Fig. 13. If altering the Fourier transform with position embedding, results of SD cannot provide texture features effectively. Meanwhile, if we do not use the RGB results of Zero123 but take its phase components, due to the lack of pixel-level supervision, the produced 3D Gaussians cannot convergence well. Consequently, the choice used in main paper is suitable and meaningful.

## H    MORE VISUAL RESULTS AND COMPARISON

To prove the superior 3D generation ability of the proposed method, we provide more quantitative results and comparisons in the form of videos in the supplementary materials. Note that for ease of browsing, we carefully craft a website. In the zip file named "Fourier123_website", there is an HTML file called "index". You can click on it and use any browser to view its contents. Part of our visual results are shown in Fig. 14, all of them can be found in the HTML file.

Moreover, the 3D Gaussians produced by our method can be extracted into meshes. Although the quality of extracted meshes is slightly degraded compared to the original 3D Gaussians, they are still exquisite. We provide some meshes in the "meshes" sub-folder of the supplementary materials, which can be visualized by existing 3D softwares such as MeshLab or Blender.

## I    LIMITATION

Due to the inherent randomness of generation methods, we share a common problem with existing 3D generation methods: occasional generation failures. It is possible to be addressed by repeated generation with different random seeds as our method only takes 1 minute to generate once.

