# OpenReview forum: "Hybrid Fourier Score Distillation for Efficient One Image to 3D Object Generation"
_ICLR.cc/2025/Conference — ICLR 2025 Conference Withdrawn Submission_

### Official Review · Reviewer_MU38 · 2024-10-29

**Soundness:** 2
**Presentation:** 3
**Contribution:** 2
**Rating:** 3
**Confidence:** 4

**Summary:**

This paper introduces a new SDS-based method to generate 3D models from single-view images.
The key idea is to incorporate both Zero123 and the original Stable Diffusion in the distillation process.
Zero123 will be in charge of generating structures in the distillation so the proposed method would use the amplitude of the predicted noise to optimize the rendering. Stable Diffusion will be in charge of generating fine details so a vanilla SDS is adopted.
Experiments demonstrate that proposed distillation on amplitude improves the results and show some better performances than baseline methods.

**Strengths:**

The main strength is that the utilization of the amplitude of Fourier transformation in SDS loss is novel. This paper claims that Zero123 produces better structures while SD produces better details. Then, the proposed method supervises renderings using only the amplitude components of the predicted noises to extract structures of generation. This idea is overall novel and new for me.

**Weaknesses:**

The main weakness is that the experiments are not convincing enough to demonstrate the effectiveness of the idea.

1. Lack of discussion on multiview generation methods. Distilling multiview diffusion models like MVDream often produce better results than those only using SD or Zero123, which is not discussed in the paper. Recent 3D generation methods are almost dominated by these multiview diffusion models and though the proposed method achieves better results, it does not show the potential to outperform the existing multiview diffusion methods.
2. The claim that Zero123 always has better structures is not well demonstrated. Only one example in Fig. 2 is shown. Even though this is validated by more examples, this observation may be strongly limited to the specific trained Zero123 model instead of being a general and fundamental attribute of any other diffusion model. Thus, the observation will be too restrictive with limited impact.
3. The results of InstantMesh and direct distillation Zero123 are not convincing. According to my experience distilling Zero123 using ThreeStudio, the results of Zero123 would be much better than the qualitative results shown in Fig. 5. I strongly suggest the authors double-check whether the implementation of the baseline method is correct or not.

**Questions:**

Refer to the weakness.

---

### Official Review · Reviewer_zap8 · 2024-11-04

**Soundness:** 1
**Presentation:** 3
**Contribution:** 2
**Rating:** 3
**Confidence:** 5

**Summary:**

This manuscript proposes an image-to-3D generation pipeline using pretrained diffusion models. The main idea is, while generic image diffusion models (SD) and finetuned novel view models (Zero-1-to-3) can be used jointly to optimize the 3D content via score distillation, SD provides better texture details while Zero-1-to-3 is 3D consistent but blurry. In order to better combine the strengths of these two diffusion models, the manuscript proposes Fourier score distillation (FSD), i.e., defining the score distillation loss on the frequency amplitudes. FSD is therefore applied to the SD model for preserving the texture details.

**Strengths:**

- Using DFT and defining a score distillation loss on the frequency amplitudes is a very original attempt at restoring texture details in the absence of accurate spatial (phase) alignment. Modeling frequency domain features is an interesting direction in generative models and I believe it's definitely worth exploring.

- The writing and clarity of the manuscript is good in general.

**Weaknesses:**

- One of the central assumptions, that novel-view models produce over-smooth results, is very outdated. This paper has only experimented with Zero-1-to-3, which was the first ever generalizable novel-view generative model. Nowadays we have Zero123++, CAT3D, and a lot of video models, some of them are open-source as well. These more recent models can already generate detailed novel views while also being more 3D consistent than Zero-1-to-3.

- While the proposed Fourier score distillation method is claimed to be able to improve the visual details, I find the qualitative results very underwhelming. All images in figure 4 look blurry and cartoonish to me. In Fig. 5, Magic123 clear has better texture details, despite some failure cases that can be attributed to weak global 3D consistency.

- Quantitatively, the evaluation metrics also cannot directly reflect the effect of the proposed FSD in improving texture details. CLIP-similarity is used for ablation studies but it is mostly reflecting the overall semantic alignment rather than texture details. The user studies also don't provide an aspect for texture details. In general, none of the experimental results can adequately support the central claim of the paper.

- There seems to be a small error with Eq 9: if z is the denoising output as defined in L231, then there should also be an Jacobian term that backpropagates the gradient from the amplitudes to the spatial domain.

**Questions:**

DFT amplitudes of a single image (or the $\epsilon$ noise) is very noisy. I'm unsure if using such a noisy gradient could really improve the score distillation quality fundamentally.

For rebuttal, please address the weaknesses listed above.

---

### Official Review · Reviewer_5Fwg · 2024-11-05

**Soundness:** 2
**Presentation:** 2
**Contribution:** 1
**Rating:** 3
**Confidence:** 5

**Summary:**

The paper proposes a method for single image to 3D named Fourier123. The method inherits from Magic123 but changes SDS loss to the frequency domain and only keeps the amplitude components. The adapted version is coined hy-FSD. It is applied on DreamFusion and DreamGaussian. It achieves better CLIP-Sim than the original SDS.

**Strengths:**

1. The method is well-motivated by insights from the frequency domain.

2. It proposes a better alternative loss for SDS, which performs better when combined with Zero123 SDS.

**Weaknesses:**

1. Low-quality texture: The appearance of 3D models looks noisy, even on colors that are supposed to be uniform. It does not look more preferable to InstantMesh, which is much faster.

2. As an optimization-based method, it is much slower than inference-only methods like Stable Fast 3D, which only takes seconds and may have better geometry and appearances.

3. Insufficient quantitative experiments: No 3D metrics are reported, such as CD, F-Score, Vol. IoU. It is only evaluated on 100 objects. Since most of the competitive baselines take no longer than a few minutes per shape, it would be proper to evaluate at least a few hundred objects on different datasets like GSO, Omni3D, etc.

**Questions:**

L522: I suggest using threestudio's implementation instead, which is a common practice and performs much faster and better than Zero123's results in the paper.

---

### Official Review · Reviewer_pfxA · 2024-11-08

**Soundness:** 3
**Presentation:** 4
**Contribution:** 3
**Rating:** 5
**Confidence:** 4

**Summary:**

The submission proposes a pipeline to generate 3D Gaussian representation from a single image using a score distillation-based method by generating high-frequency details leveraging the Fourier domain and ensuring cross-view consistency leveraging the novel view synthesis ability of Zero-1-to-3. To the review's knowledge, the perspective is new, and the results seem promising, among score distillation-based methods.

**Strengths:**

1. The exposition is very clear, and the work is comfortable to read.
2. The motivation of the approach is clear and reasonable.
3. The overall framework design seems to be natural and reasonable following the basic idea.
4. The idea is validated on two frameworks (DreamGaussian and DreamFusion), generating consistent performance improvement.

**Weaknesses:**

1. The field of 3D generation has been shifting from score distillation approaches toward feed-forward methods. The manuscript would benefit from a more comprehensive discussion of this methodological evolution and its implications for the current work.
2. While the paper mentions Wonder3D, it lacks direct experimental comparisons with this significant baseline. Additionally, Unique3D, another prominent method known for high-quality frontal appearance generation, should be included in the comparative analysis. Both Wonder3D and Unique3D have publicly available implementations, making such comparisons feasible.

**Questions:**

Do you consider applying the high-pass filter over the Fourier domain?

---

### Note · Authors · 2024-11-13

I have read and agree with the venue's withdrawal policy on behalf of myself and my co-authors.